

# Low-temperature electron mobility in doped semiconductors with high dielectric constant

Khachatur G. Nazaryan[1] and Mikhail Feigel'man[2]

**1** Department of Physics, Massachusetts Institute of Technology, Cambridge, MA 02139
**2** L.D. Landau Institute for Theoretical Physics, Chernogolovka, Russia

## Abstract

We propose and study theoretically a new mechanism of electron-impurity scattering in doped seminconductors with large dielectric constant. It is based upon the idea of *vector* character of deformations caused in the crystalline lattice by any point defects siting asymmetrically in the unit cell. In result, local lattice compression due to the elastic deformations decay as $1/r^2$ with distance from impurity. Electron scattering (due to standard deformation potential) on such defects leads to low-temperature mobility $\mu(n)$ scaling with electron density $n$ of the form $\mu(n) \propto n^{-2/3}$ that is close to experimental observations on a number of relevant materials.

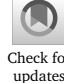

## Contents

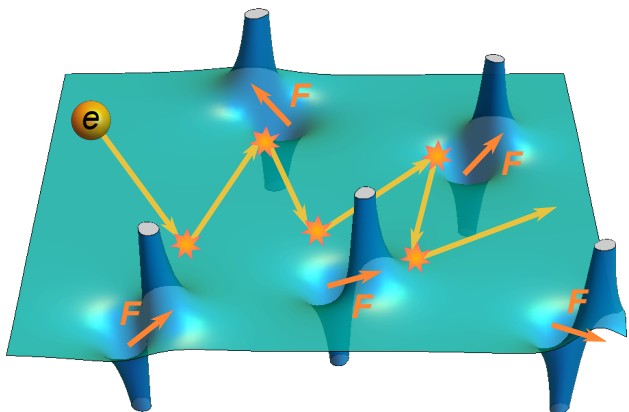

Figure 1: Schematic visualization of the proposed scattering mechanism. Electrons scatter on deformation potential induced by local vector impurities. The local vectors $F$ are oriented randomly for different impurities.

## 1 Introduction

A number of doped semiconductors is known to demonstrate low-temperature mobility $\mu(n)$ with a nearly power-law dependence on electron density, $\mu \propto n^{-\beta}$, with the exponent $\beta$ in the interval $\frac{1}{2} < \beta < 1$, see Ref. [1–6] for Strontium Titanate $SrTiO_3$, Ref. [7] for Potassium Tantalate $KTaO_3$, Ref. [1, 8, 9] for Lead Telluride PbTe and Ref. [14] for mixed-chalcogenide compound TlBiSSe. Obviously, mobility in the $T \to 0$ should be determined by impurity scattering, but it is not so easy to identify the specific mechanism of this scattering. Indeed, an obviously existing scattering by screened Coulomb potentials produced by charged impurities leads [10, 11] to $\mu_{\mathrm{Coul}}(n) \propto 1/\ln(n)$. Another omnipresent type of scattering is provided by short-range random potentials. They lead to density independent scattering cross sections $\sigma$ with mean-free path estimated as $l \propto 1/(n\sigma)$. Thus, short-range potentials yield a mobility scaling as $\mu_{\mathrm{short}}(n) \propto n^{-4/3}$. None of these mechanisms is able to explain the data [1–9]. The common feature of all these doped semiconductors is high dielectric constant of the corresponding undoped material, which makes Coulomb scattering by charged impurities very weak.

In the present manuscript we propose and study a new mechanism of electron scattering by point defects, which we call *vector impurity* mechanism. Our key idea follows from two observations: i) all considered families of semiconductors have crystal lattices with relatively complicated elementary cells, which forces lattice defects (a vacancy or a substitutional atom) to break down the symmetry of elastic media around it; as a result, such defects act as a microscopic "force" upon surrounding elastic media. ii) elastic deformations due to a point-like force $\mathbf{F}\delta(\mathbf{r})$ applied to an elastic media lead [13] to lattice deformations $\mathbf{u}(\mathbf{r})$ with slowly decaying compression $\mathrm{div}\mathbf{u} \propto 1/r^2$. Now, one can employ usual electron-phonon deformation potential Hamiltonian of the form $H_{int} \propto (\psi^\dagger\psi)\mathrm{div}\mathbf{u}$ to find that it leads to the impurity transport cross-section $v_{tr}(q) \propto 1/q^2$, with $q$ being transfered momentum. With typical $q \sim k_F \sim n^{1/3}$, one immediately find mobility $\mu(n) \propto (n v_{tr} k_F)^{-1} \propto n^{-2/3}$ which is rather close to the observations [1–9]. Below we provide detailed exposition of our approach, and apply it first to Strontium Titanate (where some complications arise due to its many-band structure), and then to $KTaO_3$, PbTe and TlBiSSe.

## 2  Elastic deformations due to vector impurities.

Conduction-band electrons in semiconductors interact with lattice distortions via deformation potential

$$\hat{H}_{\text{imp}} = D_{ac} \int d\mathbf{r} \hat{\psi}^\dagger(\mathbf{r}) \hat{\psi}(\mathbf{r}) \text{div} \mathbf{u}(\mathbf{r}), \tag{1}$$

where coupling constant $D_{ac}$ is usually rather large, about few eV. Thus we need to consider possible sources of lattice distortions leading to non-zero compression $\mathbf{u}$. In the simplest model of a "void" in isotropic elastic media, the deformations $\mathbf{u}$ which arise around it lead to $\text{div}\mathbf{u} = 0$, see [13], Problem 2 for Paragraph 7. Crucial point is to notice that any atomic defect in a complicated crystal structure will break the symmetry of the lattice in the way that is equivalent to the presence of some local **vector** source. In other words, it would be incorrect to consider Oxygen vacancy in STO just as small spherical defect in elastic media, as it would be possible in case of vacancy is simple cubic lattice with single atom per unit cell. Oxygen defects in the lattice of STO are located asymmetrically w.r.t. center of the unit cell. Thus, in terms of symmetry of elastic deformation, the effect of such a defect is equivalent to the presence of some *frozen in local force* $\mathbf{F}$. The problem of elastic deformations in the presence of such a force was first solved by W.Thomson in 1848; detailed solution is present in Ref. [13], as the Problem to the Paragraph 8. It reads as follows:

$$\mathbf{u} = \frac{1+\nu}{8\pi E(1-\nu)} \frac{(3-4\nu)\mathbf{F} + \mathbf{n(nF)}}{r}. \tag{2}$$

Here $\nu$ and $E$ are the Poisson's ratio and Young modulus respectively, the magnitude of a force $F = |\mathbf{F}|$ will serve as a fitting parameter for our theory. This force originates from the anisotropic local distortion of the lattice due to the atomic substitutions. For low enough doping concentrations these local distortions are independent from each other, and therefore the magnitude of this force is expected to be concentration independent. Local compression $\text{div}\mathbf{u}$ corresponding to deformations (2) is given by

$$\nabla \mathbf{u} = \mathcal{U} \frac{\mathbf{Fr}}{r^3}, \qquad \mathcal{U} = \frac{(1+\nu)^2}{8\pi E(1-\nu)}. \tag{3}$$

Fig. 1 presents a sketch of electron scattering on a random deformation potential caused by vector impurities.

## 3  Collision integral, relaxation time and mobility

Now we use the Hamiltonian (1) with impurity-induced compressions given by Eq. (3) to calculate electron scattering rates.

We will study electric transport in an electron system using the Boltzmann kinetic equation. To find the conductivity and the corresponding mobility within linear response theory, we expand the distribution function as $f_{\mathbf{p}} \approx n_p + \delta n_{\mathbf{p}}$. Where, $n_p = \left[\exp\left(\beta \xi_p\right) + 1\right]^{-1}$ is the Fermi-Dirac distribution with $\beta = 1/(k_B T)$ and Boltzmann constant $k_B$, and $\xi_p = E(p) - E_F$ with the Fermi energy $E_F$. Since we are concerned with the low-temperature transport we will limit our discussions for $k_B T \ll E_F$, thus in the leading order approximating the Fermi-Dirac distribution with a step function. This helps to write the Boltzmann equation in presence of electric field in the linearized form:

$$-e\mathbf{E}\mathbf{v_p} \frac{\partial n_p(\xi_{\mathbf{p}})}{\partial \xi_{\mathbf{p}}} = I\{\delta n_{\mathbf{p}}\}, \tag{4}$$

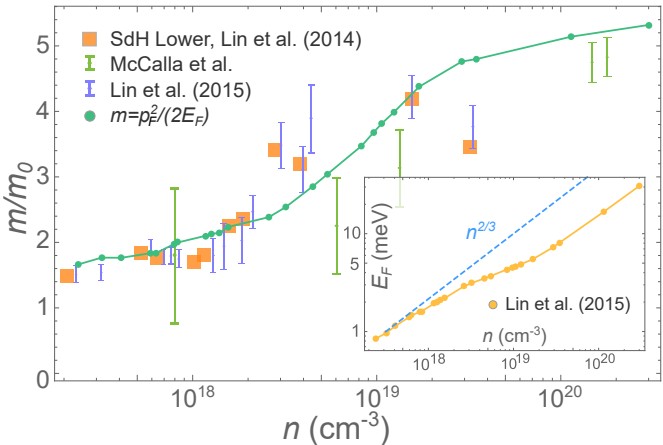

Figure 2: The effective mass $m/m_0$ (vertical) vs electron density in log scale (horizontal). Shown are the experimental values from Shubnikov-de Haas effect [19] (orange squares), from specific heat [24] (green dots), from quantum oscillations [26] (purple dots). The emerald green line showcases the $m(n)$ dependence we used, which was obtained using a model of spherical Fermi surface via Eq. (11). The concentration dependence of the Fermi energy is shown in the inset, see Ref. [26].

where $\mathbf{v_p} = \partial \xi_\mathbf{p}/\partial \mathbf{p}$ is the group velocity. The collision integral $I\{\delta n_\mathbf{p}\}$ in the RHS of the above equation describes the electron scattering at the impurity-induced compressions governed by the Eq. (3). It is explicitly expressed as follows

$$I = \frac{2\pi}{\hbar} \sum_j \int_{\mathbf{p}'} |v_{\mathbf{p}'\mathbf{p}}^{(j)}|^2 \big[\delta n_{\mathbf{p}'} - \delta n_\mathbf{p}\big] \delta\big(\varepsilon(\mathbf{p}) - \varepsilon(\mathbf{p}')\big), \tag{5}$$

where we introduced a notation $\int_{\mathbf{p}'} = \int \frac{d^3\mathbf{p}'}{(2\pi\hbar)^3}$, $\sum_j$ for summation over impurities, and $v_\mathbf{k}^{(j)}$ for the Fourier transform of the deformation potential from the Eq. (3) induced by the $j^{th}$ impurity:

$$v_\mathbf{k}^{(j)} = 4\pi i G \frac{\mathbf{F}^{(j)}\mathbf{k}}{k^2}, \quad G = \mathcal{U}D_{ac}, \tag{6}$$

where the newly defined parameter $G$ contains all the material related properties – elastic parameters and deformation potential. The deviation of the distribution function from the Fermi distribution is produced by the electric field, thus for an isotropic Fermi surface in the limit of weak electric field $E$ we can preserve only the first angular harmonics and choose $\delta n_\mathbf{p} = \mathbf{E}\mathbf{p}\frac{\partial f}{\partial \epsilon}\eta(\varepsilon)$, with the function $\eta(\varepsilon)$ being only energy dependent. As a result, the integral (5) can be evaluated explicitly, as detailed in the appendix, producing:

$$I = 2\pi \frac{mG^2}{\hbar^2}\eta(\varepsilon)\frac{\partial f}{\partial \epsilon}\sum_{\text{imp}} M_\mathbf{p}\big(\mathbf{F}^{(j)}\big), \tag{7}$$

$$M_\mathbf{p}(\mathbf{F}) = (\mathbf{E}\hat{\mathbf{p}})F^2 - 2(\mathbf{EF})(\mathbf{F}\hat{\mathbf{p}}) + 3(\mathbf{E}\hat{\mathbf{p}})(\mathbf{F}\hat{\mathbf{p}})^2, \tag{8}$$

with $\hat{\mathbf{p}}$ denoting a unit vector along the momentum $\mathbf{p}$. We emphasize that the above expression is dependent not only on the relative orientation of momentum $\mathbf{p}$ and electric field $E$, but also on the relative orientations of $\mathbf{p}$ and the vector force $\mathbf{F}$.

In order to sum over the impurities we need to average the expression (8) over the orientation of the vector $\mathbf{F}$. As a result we obtain a final expression for the collision integral in a

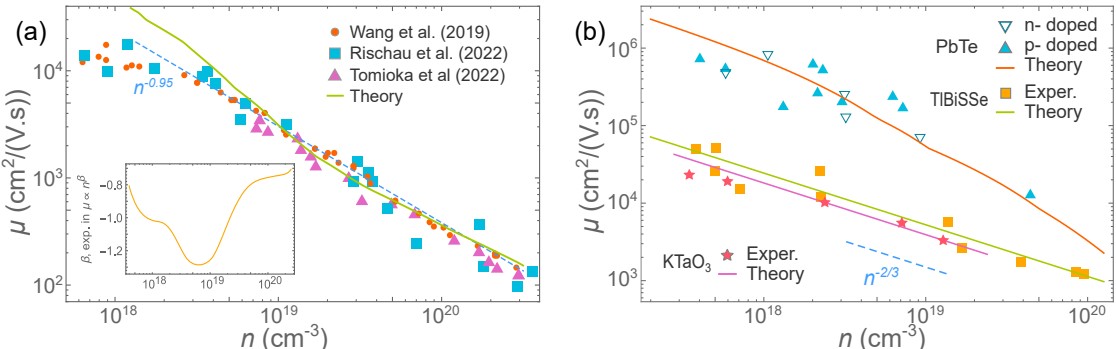

Figure 3: The mobility $\mu$ in log scale (vertical) vs electron density in log scale (horizontal). (a) The red circles, blue squares and magenta triangles correspond to the experimental data for $SrTiO_{3-x}$ extracted from [2–4] respectively, and the green line represents the theoretical model Eq. (10). The inset focuses on the local exponent $\beta = -d \ln \mu / d \ln(n)$, which varies between $-0.8$ and $-1.2$, averaging at $-0.95$; and the scaling $n^{-0.95}$ depicted with a dashed blue line. (b) Blue triangles illustrate the data for PbTe extracted from Ref. [1, 8, 9]. The red line employs our theoretical model with the effective electron mass found in [29] (see Fig. 4 in the Supplement). The orange squares and red stars indicate the mobility data for TlBiSSe [1, 14] and $KTaO_3$ [7]. The green and pink lines represent the theoretical results for electron mobility in these materials, using constant effective masses $m = 0.14 m_0$ and $m = 0.5 m_0$ respectively.

form $I = -\delta n_{\mathbf{p}}/\tau$ with a relaxation time

$$\tau = \frac{3\hbar^2 p_F}{8\pi (GF)^2 mn} . \tag{9}$$

The relaxation time is then used to find the electron mobility:

$$\mu = \frac{e\tau}{m} = \frac{3e\hbar^2 p_F}{8\pi (GF)^2 m^2 n} . \tag{10}$$

We see that for a concentration independent effective mass electron mobility scales with concentration as $\mu \sim n^{-2/3}$ as tipped off in the Introduction. However, since the mass enters squared in the above equation, even a relatively weak $n$-dependence $m(n)$ can influence the results considerably.

## 4 Application to SrTiO$_3$

One of the most interesting materials that our discussions can be applied to is Strontium Titanate $SrTiO_3$. Being a band insulator it becomes a very dilute 3D metal due to tiny doping ($10^{-6} - 10^{-3}$ conduction electrons per unit cell) and demonstrates a number of unusual properties [15–17]. They mainly originate form the close proximity of insulating STO to a ferroelectric transition, which leads to a giant low-temperature dielectric constant $\epsilon_0 \approx 20000$. As a result, Coulomb interaction in STO is strongly suppressed; the accurate consideration shows that the electron mobility produced by the scattering on Coulomb field is more than two orders of magnitude greater than the experimental data.

Having an almost spherical Fermi surface when lightly doped, at concentrations higher than $n_{c1} \sim 2 \cdot 10^{18} \mathrm{cm}^{-3}$ SrTiO$_3$ acquires a complicated multiband Fermi surface far from being

isotropic [19–23]. Anisotropic Fermi surface can potentially produce correlations between the subsequent scatterings by affecting the scattering direction probability distribution after each act of scattering. This is indeed the case for scattering on isotropic impurities where light electrons can contribute to the collision integral more dramatically than heavy ones. However, according to the Eq. (8) the collision integral depends strongly on the relative orientation of **p** and the vector force **F**, and of **E** and **F**. Since these vector forces are oriented randomly, the exact shape of the Fermi surface does not seem to be relevant and the electron scattering is effectively averaged out. This enables us to model the electron dynamics with a spherical Fermi surface with an effective mass introduced phenomenologically as

$$m = \frac{p_F^2}{2E_F} = \frac{\hbar^2 \left(3\pi^2 n\right)^{2/3}}{2E_F}, \tag{11}$$

with the Fermi energy obtained from the experimental data. Somewhat similar approach is used in [24]. Fig. 2 summarizes a number of experimental data for the effective mass in the lowest band of STO, obtained by different kinds of experiments: Shubnikov-de Haas effect, quantum oscillations and the density of states (DoS) mass found from specific heat measurements. Continuous green line in the same plot shows the dependence $m(n)/m_0$ which we extracted using the data from Ref. [26] for Fermi energy and Eq. (11), to be used in our further calculations.

The above considerations allow us to directly implement the result given by Eq. (10) for the analysis of experimental data on SrTiO₃. Fig. 3a) compares our theoretical results for electron mobility with the experimental data [2–4]. We used here single fit parameter, the strength of vector impurity potential $F$. The results are in a good agreement with the experiment showing a reasonable overall scaling with deviations not exceeding 10% for $n > 5 \cdot 10^{18}$ cm$^{-3}$.

Our approach improperly predicts the mobility behaviour at lowest concentrations where experimental data demonstrate saturation of $\mu(n)$ with further decrease of $n$ below $n_c \sim 5 \cdot 10^{18}$ cm$^{-3}$, which is not described by Eq. (10). It means that another scattering process should be taken into account to describe this feature. One possible effect could come from Coulomb interaction which leads to slow logarithmic dependence of $\mu(n)$. However it is easy to check that Coulomb scattering itself would lead to mobility overestimated by 2 orders of magnitude. Another possible explanation could be electron scattering on domain walls. These processes can be roughly modeled using a relaxation time defined as $\tau = l/v_F$, where $l$ is the characteristic domain size. In order to fit the experimental data for STO this approach requires the domain size to be $l \sim 0.5\mu$m, wheres the experiment [37] reveals the domain size to be an order of magnitude larger. Finally, we would like to mention spatial non-uniformity of dopant's concentration as a possible source of $\mu(n)$ saturation at lowest $n$; we leave investigation of this issue for future research.

Now we need to implement "sanity check" to see how large are the lattice deformations induced by our vector impurities. Let us evaluate the characteristic displacement $u(a)$ at the minimal distance from the impurity, using known parameters of STO, like the deformation potential $D_{ac} \approx 4$ eV [27], Young modulus $E \approx 270$ GPa and Poisson ratio $\nu = 0.24$ [28]. To describe the experimental data we used a fit parameter $F \approx 9.1 \cdot 10^{-9}$ N. According to Eq. (2), it corresponds to largest atomic displacement

$$\frac{u(a)}{a} \approx 3\%, \tag{12}$$

where $a = 0.39$ nm is the STO lattice constant. Such a maximal displacement does not seem to be unreasonable.

# 5 Application to other materials

Now we extend our analysis for several other doped semiconductors with high dielectric constants. Namely, we use our approach to describe electron mobility in a wide-gap semiconductor perovskite Potassium Tantalate $KTaO_3$, Lead Telluride PbTe - narrow gap semicoonductor, and a zero-gap semiconductor - mixed-chalcogenide compound TlBiSSe; their dielectric constants are roughly 4500, 1000 and 20 respectively.

Effective electron mass in PbTe depends on the electron density substantially [29], increasing from $0.07m_0$ at $n = 2 \cdot 10^{17}$ cm$^{-3}$ up to $0.5m_0$ at $n = 10^{20}$ cm$^{-3}$. The corresponding data from Ref. [29] are illustrated in Fig. 4 in the Supplement for convenience. We employed interpolation of these actual experimental data for the calculation of the mobility dependence $\mu(n)$ in PbTe within our theory. Concerning effective masses for $KTaO_3$ and TlBiSSe, we are not aware of any data for $m(n)$ dependencies, therefore we used the following constant values for these masses: $m = 0.5m_0$ [7] and $m = 0.14m_0$ [1] respectively.

To calculate mobility $\mu(n)$ dependence according to our theoretical formula (10), we need to use the data for the deformation potential $D_{ac}$, Young modulus $E$ and Poisson ratio $\nu$, see Eqs.(6) and (3). For $KTaO_3$ we used $E = 215$ GPa and $\nu = 0.24$, see Ref. [31, 32]. We did not find data for the KTO deformation potential and thus used, for general orientation, the value $D_{ac} = 4$eV known for STO, as these materials are rather similar. For PbTe we used $E = 57.5$ GPa and $\nu = 0.26$, see Ref. [33], and deformation potential $D_{ac} = 15$ eV, see Ref. [34, 35].

With the material parameters mentioned above, we are left with just single unknown parameter $F$, the magnitude of "vector force" related to impurities in KTO and PbTe. We fit the values of this parameter to obtain best agreement between our theory and the data, the results are shown in Fig. 3b). The overall agreement is clearly rather good, supporting the ubiquity of the proposed mechanism.

Using the values of $F$ equal from the fit, namely $F = 2.6 \cdot 10^{-9}$ N for KTO and $F = 5.8 \cdot 10^{-10}$ N for PbTe, we estimate the analogues of Eq. (12), the largest relative lattice displacements $u(a)/a$ due to vector impurities. We found $u(a)/a \approx 5.5\%$ for KTO and $u(a)/a \approx 0.4\%$ for PbTe. In addition, we present in Fig. 3b) the best fit for the $\mu(n)$ dependence in TlBiSSe. In this case we did not found the data for deformation potential and elastic modulus, thus we used for the fit the whole coefficient in front of $n^{-2/3}$ dependence.

# 6 Conclusions

We developed a new theory of electron - impurity scattering in low-electron-density materials with high dielectric constant. Low electron density makes it possible to vary it in a broad range, by few orders of magnitude. The observed in many materials dependence of low-temperature mobility on density, $\mu(n)$, could not find any explanation in terms of scattering on Coulomb or short-range potentials. The notion of vector impurities we propose in this manuscript helps to elucidate the origin of unusual type of scattering due to slow-decaying deformation potential.

In its simplest form, our theory predicts $\mu(n) \propto n^{-2/3}$ which is not far from the data on several low-density materials. Moreover, the account of the density-dependence effective mass $m(n)$ allows us to obtain theoretical results in a very good agreement with the data. These results are provided in Fig. 3a) for the case of Oxygen-deficient Strontium Titanate, and in Fig. 3b) for several other semiconductors: $KTaO_3$, PbTe and TlBiSSe.

Still an open issue for our theory is related to Nb-substituted Strontium Titanate which demonstrate similar $\mu(n)$ dependence at low temperatures: in this case it is not clear why substitution of Sr atom by Nb produces vector impurity. We leave this problem for future studies.

## Acknowledgements

We are grateful to Kamran Behnia and Mikhail Glazov for many useful discussions. This research was supported by the Russian Science Foundation Grant No. 21-12-00104.

## A  Collision Integral

Here we present the detailed evaluation of the collision integral, Eq. (5) in the main text. First, we carry out a Fourier transformation for the potential expressed through the Eq. (6)

$$
v_{\mathbf{k}} = 2\pi G \int_0^\infty dr \int_0^\pi F_\parallel \cos\theta\, e^{ikr\cos\theta} \sin\theta\, d\theta = 2\pi F_\parallel \int_0^\infty dr \int_{-1}^1 t e^{ikrt} dt
$$
$$
= 4\pi Gi \frac{(\mathbf{F}, \mathbf{k})}{k^2}\,. \tag{A.1}
$$

Let us consider $\mathbf{k} = \frac{1}{\hbar}(\mathbf{p} - \mathbf{p}')$ with the $z$ axis oriented along $\mathbf{p}$ and for sake of simplicity we recall that the investigated scattering process conserves energy, thus $|\mathbf{p}| = |\mathbf{p}'|$.

$$
v_{\mathbf{p}'\mathbf{p}} = 4\pi Gi\hbar \frac{(\mathbf{F}, \mathbf{p} - \mathbf{p}')}{|\mathbf{p} - \mathbf{p}'|^2} = 4\pi Gi\hbar \frac{F_\parallel (1 - \cos\theta) - F_x \sin\theta\cos\phi - F_y \sin\theta\sin\phi}{2p(1 - \cos\theta)}\,, \tag{A.2}
$$

here the angles $\theta, \phi$ denote the orientation of the vector $\mathbf{p}'$. Since we still have a freedom of orienting $x, y$ axes, we can $F_y = 0$:

$$
v_{\mathbf{p}'\mathbf{p}} = \frac{2\pi Gi\hbar}{p}\left(F_\parallel - \frac{F_x \sin\theta\cos\phi}{(1 - \cos\theta)}\right)\,. \tag{A.3}
$$

This expression is then plugged into the collision integral, yielding

$$
I_{\text{imp}} = -\frac{2\pi G^2}{\hbar}\left(\frac{\partial f}{\partial\epsilon}\eta(\epsilon)\right)\sum_{\text{imp}}\int\left(F_\parallel - \frac{F_x \sin\theta\cos\phi}{(1 - \cos\theta)}\right)^2
$$
$$
\times\left[E_\parallel(1 - \cos\theta) - E_x \sin\theta\cos\phi - E_y \sin\theta\sin\phi\right]\cdot\frac{m\sin\theta\, d\theta\, d\phi}{2\pi\hbar}\,, \tag{A.4}
$$

where we have already carried out the trivial integration of the $\delta$ function. The further integration over the angles leads to the equation (8) from the Main text.

## B  Effective Electron Mass in PbTe

As discussed in the main text, the effective electron mass in PbTe depends on the electron density considerably [29]: upon increasing concentration from $n = 2\cdot 10^{17}$ cm$^{-3}$ to $n = 10^{20}$ cm$^{-3}$ the electron mass enhances from $0.07m_0$ up to $0.5m_0$. In Fig. 4 we present the experimental data from Ref. [29] as well as the interpolating function $m(n)$ which we used to evaluate the mobility.

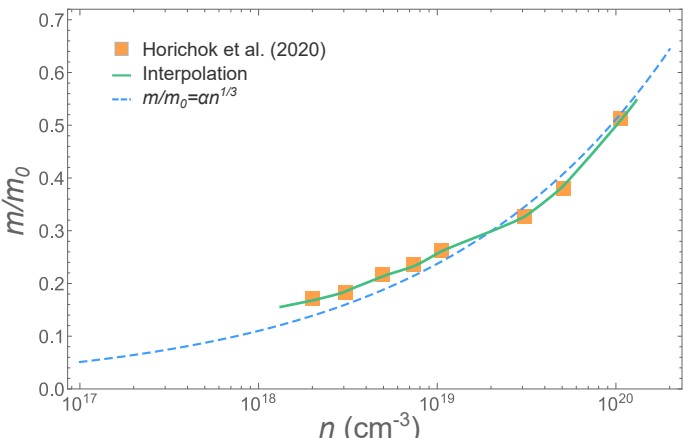

Figure 4: The effective mass of electrons as a function of the concentration for PbTe. The orange squares showcase the experimental data [29], the green line illustrates our interpolation employed in further calculations and the dashed blue line approximates tha data with a scaling $m/m_0 \sim \alpha n^{1/3}$ with $\alpha \approx 1.1 \cdot 10^{-7}$.

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
