# Peer review of "Low-temperature electron mobility in doped semiconductors with high dielectric constant"

_SciPost Physics, doi:SciPost Phys. 14, 046 (2023)_

## Round 1 · Referee Report · Anonymous (Referee 1) · 2022-10-12

Strengths
1-Originality
2-Timeliness
3- Relevance to experimental data
Weaknesses
1- The physical meaning of the force F is barely discussed.
2- Comparison with the experimental data is somewhat selective and the discrepancies not sufficiently presented.
Report
See the attached file.
Requested changes
See the attached.

---

## Round 1 · Referee Report · Anonymous (Referee 2) · 2022-10-17

Report
In the manuscript, the electron mobility dependence on impurity concentration in doped semiconductors with high dielectric constant is examined. The aim of this work is to explain the power-law dependence of the mobility observed in a number of recent experiments. Due to large dielectric constants of the studied materials, the Coulomb impurity scattering is irrelevant, and some other scattering mechanism should dominate. According to the experimental data, this mechanism results in the correct concentration dependence if the scattering potential behaves as 1/r^2. The author’s proposal is that such a potential can be caused by a deformation induced by the point-like defects asymmetrically located in the elementary cell. The latter is large and complicated in the studied materials, which supports the author’s idea.
The proposed idea is formulated in Eqs. (1)-(3). Then, in Eqs. (4)-(8), authors make standard textbook transformations which can be found, e.g., in [A.A. Abrikosov, Fundamentals of the theory of metals, 1988. United States: Elsevier Science Pub Co Inc.] and obtain the transport relaxation time and the mobility. The scattering potential ~1/r^2 does not allow for description of the experimental data, and the authors take into account that, not only the transport relaxation time, but also the effective mass of carriers depends on the concentration, as it follows from independent measurements. The effective mass varies up to the factor of 5 according to Fig. 1. Account for this strong mass variation allows for explaining the mobility dependence on the concentration for a few materials in a wide concentration window, Fig. 3. The authors also check that the fitting parameter corresponds to reasonable deformation values.
The manuscript looks scientific and convincing, and it can be published after clarifying the following points.
1. If there are some independent witnesses for the defects asymmetrically positioned in the unit cells? Is it possible to find them by, e.g., AFM technique? If there are relevant publications on this issue?
2. Why do these scatterers not scatter electrons like defects with a short-range potential? The authors exclude this mechanism because it results in an incorrect dependence of the mobility, but why this it is excluded physically? It should be much stronger than the proposed deformation-potential mechanism requiring an asymmetrical position of a defect in the unit cell.
3. What is the reason for the effective mass dependence on concentration? This strong dependence means a sufficient nonparabolicity of the energy spectrum. In this case the density-of-states mass is defined as m=p/(dE/dp). How does it correspond to Eq. (11)?

---

## Round 2 · Referee Report · Anonymous (Referee 2) · 2022-11-2

Report

The work can be accepted in its present form

---

## Round 2 · Referee Report · Anonymous (Referee 1) · 2022-11-7

Report

The authors have satisfactorily addressed all my questions. I recommend publication.

---

## Round 2 · Author Response

Dear Dr Attaccalite,

Thank you very much for arranging reviews of our manuscript “Low-temperature electron mobility in doped semiconductors with high dielectric constant” and thank you to reviewers for a careful appraisal of the paper. We are delighted to see that they like our work. Please find herewith the manuscript revised according to their recommendations. We agree with reviewers’ suggestions and implemented them in the revised manuscript as described below.

respectfully, authors

Point to point reply to referees’ comments

Referee 1

  1. In the introduction, they write: “Another omnipresent type of scattering is provided by short-range random potentials, but this one leads to density - independent scattering cross-section, thus $\mu_{short} \propto n^{-4/3}$. Why? As far as I see, if there is a random distribution of potential wells with a density of n, the average distance in three dimensions and the mean-free-path would be $\propto n^{-1/3}$ and therefore $\mu_{short} \propto l/k_F \propto n^{-2/3}$. Would the authors explain?

Author comments: The referee provides a simple derivation for the mobility leading to a scaling $n^{-2/3}$. Evaluating the mean-free path of electron with the average distance between atoms $l_0 \propto n^{-1/3}$, the scattering time is then expressed as $\tau \sim l/v_F$ giving the above mentioned scaling for mobility $\mu = e\tau /m$. This derivation however assumes that in average electrons should definitely experience scattering after traveling distance $l_0$ (average distance between impurities). This is not true for short range potentials with cross section $\sigma$ much smaller than $l_0^2$. The more accurate way for estimating the mean-free path $l$ is now described in the manuscript and is achieved by requiring that there is at least one scattering center in the tube of length $l$ and cross section $\sigma$, i.e. $n l \sigma \sim 1$. For short-range potentials the scattering cross-section is energy independent (e.g., for an impenetrable ball of radius $a$, the cross section can be semi-classically estimated as $4\pi a^2$), and therefore the resulting mean-free path scales as $l\propto 1/n$, leading to the dependence stated in the manuscript for short-range potentials.

  1. Equation 10 yields an expression for mobility, including the parameter GF. G, has the dimension of volume and F is a force. The authors do not discuss the physical meanings of these two parameters. Presumably G is set by elastic properties of the crystal (Young Modulus and the Poisson ratio) and F is the elastic force generated by the impurity. The authors use F as a fitting parameter. Why should F be independent of impurity concentration? If F varies just a little with n then the postulated μ∝n^(-2/3) will no more be true.

Author comments: Firstly, the referee claims that the physical origin of the parameter $G$ is not elaborated properly. We believe there is some clear misunderstanding here, since the parameter $G$ is introduced in the Eq. (6) of the main text as the product of the deformation parameter $D_{ac}$ and parameter $\mathcal{U}$. The explicit expression for the latter is provided in the Eq. (3) through elastic coefficients (Young Modulus and the Poisson ratio). Next, the referee asks about the possible n-dependence of the vector force F which serves as a fitting parameter of our problem. This force originates from the local distortion of the lattice due to the atomic substitutions. Since the doping concentrations are very low, these local distortions are independent from each other. Therefore, we don’t see reasons that could lead to the force $F$ becoming concentration dependent. We implemented the above discussions into the main text.

3-4. I think the authors should tell their readers that in the case of strontium titanate, two experimental groups find that 𝜇 ∝ 𝑛 −𝛼 with 𝛼 ≅ 5/6 in both Nb-doped strontium titanate (Tomioka et al. arXiv:2203.16208) and in oxygen-reduced strontium titanate (Wang et al. npj Quantum Materials 4 : 61 (2019)). In the latter study, it was also found that when STO is doped with Ca, α decreases from 0.84 to 0.58. Therefore, the experimental data does confirm that the mobility has a power law dependence, and the carefully quantified exponent is close to 0.67, but different from it.

It is also worth noticing that the variation of the exponent with Ca content indicates a role played by the dielectric constant (which varies when Ca is introduced). Now, dielectric constant of the material is absent in Equation 10. On the other hand, it does play a role in the picture drawn in reference 1.

Author comments: Thank you for your suggestion, we added the data from the paper Tomioka et al. arXiv:2203.16208 in the Fig. 3(a) of the revised manuscript. The referee addressed the phenomenon of Ca doping resulting in a decline of the scaling exponent from 0.84 to 0.58. The referee notes that the Ca doping affects the dielectric constant, while our final equation does not contain dielectric constant. However, Ca doping gives rise to local dipole moments. These dipoles then serve as additional sources of scattering, directly affecting electron mobility. This is an entirely different mechanism and, therefore, the experimental data in the presence of Ca-doping cannot be described based only on our theory.

  1. The authors write that “a vacancy or a substitutional atom … breaks down the symmetry of elastic media around it”. In the case of ABO3 perovskytes, this is true of oxygen vacancies, but not of A-site or B-site substitution at least if the impurities do not interact with each other. This brings us back to the meaning of F and its concentration dependence. (See ii). One Nb atom will not break the local symmetry, but two Nb atoms will create a distortion along the segment connecting them.

Author comments: We agree with the points raised by the referee’s. We discuss this question in the paper. Indeed, still an open issue for our theory is related to Nb-substituted Strontium Titanate which demonstrate similar $\mu(n)$ dependence at low temperatures: in this case it is not clear why substitution of Sr atom by Nb produces vector impurity. We leave this problem for future studies.

Referee 2

  1. If there are some independent witnesses for the defects asymmetrically positioned in the unit cells? Is it possible to find them by, e.g., AFM technique? If there are relevant publications on this issue?

Author comments: Crystal structure of perovskite is well-known and it follows from it that Oxygen vacancy breaks the symmetry of the unit cell. On the other hand, it does not seem possible to use AFM for the bulk material we discuss here.

  1. Why do these scatterers not scatter electrons like defects with a short-range potential? The authors exclude this mechanism because it results in an incorrect dependence of the mobility, but why is it excluded physically? It should be much stronger than the proposed deformation-potential mechanism requiring an asymmetrical position of a defect in the unit cell.

Author comments: In our view there is a chicken and egg problem here. The referee restates our story saying that we exclude scatterings on short-range potential defects which can potentially be more important in the transport problem than the novel mechanism developed in the manuscript. Yet, the key message of our paper is a very different one. Namely, our paper looks for a scattering mechanism that can describe the unusual scaling of the mobility of the studied materials. Since, the short-range potential yields a scaling $n^{-4/3}$ which has nothing to do with the experimental data, we had to find a scattering source of another kind. In this matter, the question of the referee seems to ask, why there is no strong enough short-range potential in this material to dominate the scattering process? This is one of the intrinsic properties of this particular material(s), and is laid in its nature. Otherwise, it would demonstrate the scaling $n^{-4/3}$, characteristic to short-range potential scattering.

  1. What is the reason for the effective mass dependence on concentration? This strong dependence means a sufficient nonparabolicity of the energy spectrum. In this case the density-of-states mass is defined as m=p/(dE/dp). How does it correspond to Eq. (11)?

Author comments: Relation m=p/(dE/dp) proposed by the referee is one of possible phenomenological relations which can be used for approximate description of non-parabolic spectrum; we used slightly different description, Eq. (11). Its major difference from the relation m=p/(dE/dp) is that our relation (11), being an integral one, is better suited for the case of strongly anisotropic spectrum known for STO at moderate concentrations. The main text pays much importance to clarifying the reasons why such an approach is reasonable in the discussed context. Indeed, anisotropic Fermi surface can potentially produce correlations between the subsequent scatterings. However, we see that, according to the Eqs.(7),(8) in the main text, the collision integral depends strongly on the relative orientation of momentum p and the vector force F. Since the latter is oriented randomly for each scattering center, the non-parabolic spectrum is effectively averaged leading to an isotropic Fermi surface with the mass determined through the Eq. (11).

---

## Round 2 · List of Changes

Implemented suggestions of the referee 1:

1) Implemented a more detailed discussion on short range potentials. 2) Elaborated the physical meanings of the parameter G and local force F. 3) Added the experimental data from Tomioka et al. arXiv:2203.16208 in the Fig. 3(a). 4) Discussed the question of the Nb doping case.

Implemented suggestions of the referee 2: 1) Discussed why the short range potentials are not important in the investigated problem. 2) Justified the approach used to introduce the effective mass.

---

## Editorial Decision

published